# The Many Faces of Gene Regulation in Cancer: A Computational Oncogenomics Outlook

**DOI:** 10.3390/genes10110865

**Published:** 2019-10-30

**Authors:** Enrique Hernández-Lemus, Helena Reyes-Gopar, Jesús Espinal-Enríquez, Soledad Ochoa

**Affiliations:** 1Computational Genomics Division, National Institute of Genomic Medicine, Mexico City 14610, Mexico; hreyes.gopar@gmail.com (H.R.-G.); jespinal@inmegen.gob.mx (J.E.-E.); mochoa@inmegen.edu.mx (S.O.); 2Centro de Ciencias de la Complejidad, Universidad Nacional Autónoma de México, Mexico City 04510, Mexico

**Keywords:** computational oncogenomics, gene expression regulation, multi-omics, integrative biology

## Abstract

Cancer is a complex disease at many different levels. The molecular phenomenology of cancer is also quite rich. The mutational and genomic origins of cancer and their downstream effects on processes such as the reprogramming of the gene regulatory control and the molecular pathways depending on such control have been recognized as central to the characterization of the disease. More important though is the understanding of their causes, prognosis, and therapeutics. There is a multitude of factors associated with anomalous control of gene expression in cancer. Many of these factors are now amenable to be studied comprehensively by means of experiments based on diverse omic technologies. However, characterizing each dimension of the phenomenon individually has proven to fall short in presenting a clear picture of expression regulation as a whole. In this review article, we discuss some of the more relevant factors affecting gene expression control both, under normal conditions and in tumor settings. We describe the different omic approaches that we can use as well as the computational genomic analysis needed to track down these factors. Then we present theoretical and computational frameworks developed to integrate the amount of diverse information provided by such single-omic analyses. We contextualize this within a systems biology-based multi-omic regulation setting, aimed at better understanding the complex interplay of gene expression deregulation in cancer.

## 1. Cancer as a Complex Phenotype

Cancer is a complex disease characterized by a series of highly variable and inhomogeneous phenomena. The set of individual and environmental factors associated with the onset and progression of cancer is large and diverse. These factors include several types of DNA mutations, chemical modifications to the DNA and histone proteins leading to epigenomic changes, alterations to the three-dimensional structure of chromatin and various processes that introduce biases resulting in clonal and sub-clonal selection. These changes impact the different levels of the gene regulatory programs, thus modifying gene expression in cancer. Gene regulatory processes are additionally affected by changes in the metabolic and signaling activity of the tumor cell and its surrounding microenvironment [1,2,3,4].

The rise of high-throughput omic technologies has improved our understanding of the complex regulatory landscape in both normal and tumor cells. The methodologies provided by the development of these technologies have become paramount for oncological research in the basic and clinical settings, thus paving the way to translational personalized medicine [5]. However, the high amount of data provided by these methods needs to be supplemented with appropriate analysis and integration schemes if they are to provide insightful theories and more importantly, successful diagnostic, prognostic and therapeutic tools.

Currently, there is a need for computational implementations to handle the analysis of data created by the ever-growing arsenal of omic technologies. Such tools should allow detailed quantitative descriptions of complex and noisy data, which in turn, ought to set the foundations for integrative modeling approaches. For these reasons, an upcoming field in computational biology termed computational Oncology has attracted interest from the scientific community [6,7,8]. Broadly, computational oncology is considered to include two main research branches; one of them focused on data processing tasks is usually called cancer bioinformatics, and the other one, cancer systems biology or systems oncology, aims at translating a large amount of analyzed data into some form of rational model that can be used to drive focused experimental research with the potential of being useful in the clinical context [9,10,11,12,13].

Cancer is widely known that is a gene-based disease, mutations in specific sequences of crucial genes, such as *p53*, *MDM2*, *BRCA1*, and *BRCA2*, to mention a few, are positively correlated with cancer appearance, development, and prognosis. However, mutations are not the only mechanism behind cancer appearance. The whole gene expression pattern, which is in turn affected by other factors is responsible for the oncogenic phenotypes.

The present review discusses the current state of the computational oncology field. In the next section (Section 2) we discuss in detail both, the well-established and emerging features that have been associated with deregulated gene expression in cancer. These features and the processes from which they arise are usually measured and analyzed separately, giving rise to individual (even fragmented) results and ensuing a relatively incomplete portrait of the complex phenotype, thus highlighting the multifactorial character of genomic regulation in cancer. Section 3 introduces the omic techniques currently used to study different genomic aspects known to impact gene regulation, how they can be used to probe tumor cell behavior and which computational methods are available to analyze their data output. Finally, in Section 4 we discuss the need to develop data integration frameworks and multi-dimensional models to account, not just for the individual contributions of each omic/regulatory layer but also for their concerted interplay and its contribution to shaping altered gene regulatory patterns.

## 2. The Multifactorial Character of Gene Regulation in Cancer

A central issue towards the molecular understanding of tumorigenesis is the progressive increase of altered gene expression patterns as a consequence of the aberrant genetic and epigenetic programs. Many elements are known to participate in the regulatory mechanisms of gene expression (see Figure 1). Some of them have been studied for decades and their operation is substantially understood, others await to be fully characterized. As it turns out, a myriad of events affect gene regulation in cancer, from direct changes in the DNA sequence (single nucleotide variants, insertions/deletions, inversions, translocations, copy number variation, gene fusions, among others) impacting gene sequences, transcription factor (TF) genes or regulatory sequences, to post-translational and epigenomic alterations.

The synthesis of RNA by RNA polymerase enzymes is termed transcription and it is responsible for the timely expression of the required genes and therefore proteins that ensure cells perform their functions properly. Because transcription must be tightly regulated to be specific, yet flexible, it is exquisitely controlled by a set of interconnected mechanisms comprising specific DNA sequences “cis elements” and specialized proteins “trans elements” that operate in the nuclear epigenetic context to activate and repress genes in response to stimuli (i.e., signal transduction) or to manage developmental and differentiation processes.

Mutations that disrupt either cis-elements or the gene sequence of trans-elements involved in transcription can lead to dangerous deregulation of gene expression, which could eventually result in cancer [14,15]. The latter has been evidenced by the fact that non-coding mutations in cancer can have regulatory impact [16,17] and that several oncogenes are actually TFs [18]. It has also been studied how cancer cells repurpose the transcriptional mechanisms to ensure proliferation [19] and dissemination [20].

Additionally, a number of factors known to have an influence on the regulation of gene expression on the rest of this section we will present some preliminaries on most of these factors.

### 2.1. The Role of Promoters and Enhancers

Promoter and enhancer cooperativity (P+E) is an established mechanism of gene expression regulation. Promoters are DNA sequences located near the transcription start site of genes comprising specific sequences known as response elements that are used as binding sites for the transcriptional machinery. On the other hand, enhancers are DNA sequences capable of binding activator proteins that recruit the transcriptional machinery to distant promoters. Enhancers are regulatory elements that can act as key regulators of tissue-specific gene expression. Promoter regions often work in conjunction with specific enhancers. A role for P+E in the expression changes associated with carcinogenesis and tumor sustenance has been established [17].

It has also been reported that mutations in cancer can lead to enhancer misregulation. For example, it has been extensively shown the crucial role that MLL3/MLL4/COMPASS-like family of histone H3 lysine 4 (H3K4) monomethyltransfereases have in cancer [21,22] and also the role that TERT promoter mutations leading to upregulation of telomerase expression observed in human cancer [23,24]. Thus, P+E activity has been recently considered a promising option in cancer therapeutics [25].

### 2.2. The Effect of Chromatin Structure

Structurally different regions in the chromatin fiber are able to control gene expression to a certain extent [26,27]. Traditionally and at a coarse-grained scale, genomic regions are distinguished according to whether they are in an open chromatin configuration (euchromatin) or in a closed chromatin state (heterochromatin). The former allows TFs and the transcriptional machinery to bind regulatory DNA sequences, while the latter precludes gene expression [28]. Changes in the euchromatin/heterochromatin distributions have been investigated and linked to cancer development [29].

### 2.3. DNA Methylation and Other Chemical Modifications

Chemical modifications to the DNA molecule, namely the covalent attachment of methyl groups to cytosine nucleotides, are able to modulate transcription at different levels: from fully preventing the process, through modulating it, to activating transcription. Differential methylation also influences spatial chromosomal configuration [30] and some DNA methylation patterns are so pervasive in some cancers [31,32,33,34], to the point where they have been used as biomarkers for classification and prognosis [35,36]. On the other hand, the distribution and intensity of chemical modification at the nucleosome level, including the covalent attachment of methyl, acetyl or phosphoryl groups and the addition of ubiquitin or SUMO proteins to histone N-terminal tails contributes to the already complex phenomenon of gene regulation in cancer.

### 2.4. Post-Transcriptional Processing

Once a primary mRNA has been synthesized by the transcriptional machinery, it is often subject to cleaving processes in which specific exons and introns are discarded or included in a final transcript in order to generate function-specific versions of mRNAs. This post-transcriptional processing is called splicing and variations in the way it is carried out are known to be cancer-related [37]. This is explained by the fact that splicing variants give rise to different proteins that display different, sometimes even antagonistic, biomolecular behaviors [38]. Therefore, splicing variants in cancer are finding their way into clinical interventions [39].

### 2.5. Chromosome Aberrations and Chromosomal Instability

Aneuploidy, is a chromosome instability alteration characterized by amplification or deletion of entire chromosomes or chromosomal sections and it has been found to be frequent in cancer [40,41,42,43]. Such enormous alterations to genes ratios are inevitably associated with gene expression modifications [44,45]. For instance, the region Chr8q24.3 suffers amplifications in different tumors [32,46,47,48] and it has also been reported in connection to oncogene overexpression and poor prognosis [49,50,51]. Specific regions have been described as significantly altered in different breast cancer subtypes, exemplified by the work of Smid and coworkers [51] who were able to characterize 313 primary breast tumors by their chromosomal instabilities.

### 2.6. Non-Coding RNAs

Non-coding RNAs (ncRNAs) are gaining increasing attention as central players behind transcriptional regulation. A decade-long research program has supported the role of ncRNAs as key drivers in cancer [52]. There are at least two relevant classes of regulatory ncRNAs and their main characteristics are their length: long non-coding RNAs are transcripts longer than 200 nucleotides, while micro-RNAs (miRs) are around 22 nucleotides long.

The phenotype-specific regulatory abilities of lncRNAs have been proven [53] and their specific role in several types of cancer has been extensively investigated. For instance, MALAT1, NEAT1, LED, HOTAIR, and MEG3, all of which are lncRNAs have been reported to target cancer-specific pathways [54,55,56,57,58,59]. miRs are smaller molecules that serve as posttranscriptional gene regulators. To date, there is a large collection of studies supporting the potentially oncogenic and tumor-suppressive c roles of miRs [60,61,62,63,64,65] regulating oncogenes or tumor suppressor genes. Given that their activity tends to be context-dependent, miRs involved in cancer gene regulation are termed oncomirs [62,63,66], regardless of whether their activity is pro- or anti-tumoral. Resorting to miR-gene regulatory networks, work from our own group [67] has shown that overexpression of members of the miR-200 family triggers a switch controlling epithelial-to-mesenchymal transition in breast cancer, via downregulation of VIM, ZEB1 and ZEB2, accompanied by aberrant TGFB signaling controlled by overexpression of the miR-199 family.

### 2.7. RNA Stability and Transport

Once the mRNA transcript has been synthesized and preprocessed by splicing in the cell nucleus, it must be transported through the nuclear membrane to the cytoplasm where translation can be initiated at the ribosomes. The import and export of molecules from the cell nucleus are controlled by a family of nucleoporin proteins that assemble at the nuclear membrane to form nuclear pore complexes. mRNA transcripts associate with nuclear transport proteins called exportins to achieve export from the nucleus through nuclear pore complexes. The process of mRNA export is regulated by the stability of the transcripts, both in their free and bound-to-exportins form and by the kinetics of the associated transport process [68,69]. These processes are known to be anomalous throughout cancer establishment and maintenance [70,71], and some of the involved molecules have been proposed as cancer biomarkers and therapeutic targets [72,73,74].

Aside from these well know features, there are other genomic factors that are now beginning to be analyzed in connection to gene expression regulatory programs. Some of these were already considered to participate in regulating gene expression, however, it was until the availability of trustworthy experimental techniques and computational methods that their unequivocal contribution to regulation was established. In this subsection, we address some of the most relevant and novel elements in gene regulation and their connection to cancer biology.

### 2.8. Chromothripsis and Other Catastrophic Chromosomal Events

Chromothripsis is another cancer-associated molecular phenotype in which several chromosomal rearrangements occur simultaneously in localized regions [75,76]. Particular DNA bridges (termed micronuclei) may be related to damage of the nuclear envelope leading to these broad chromosomal rearrangements [77,78]. Chromothripsis is often concomitant with other complex chromosomal events, such as aneuploidies [79,80], localized regions of hypermutation (kataegis) [81,82] or chromoplexy (abundant DNA translocations and deletions that appear independently in multiple chromosomes) [83,84]. The co-existence of such abnormal chromosome architecture may obey a common origin. Initially, it was considered that chromothripsis appears in approximately 2–3% of the cancer cases [75], though recent studies argue that chromothripsis may be present in up to 50% of cases in some cancer types [85].

### 2.9. 3D Structure and DNA-Associated Complexes

Epigenomic modulation is key to regulating gene expression. Aside from chemical modifications to DNA and histones or local chromatin accessibility, higher-order chromatin structures have also resulted to be associated with transcriptional regulatory processes. Recently, it was discovered that other functional molecules involved in the epigenomic control of gene expression are SATB1, CTCF, and cohesin. These proteins are known to be relevant in tumor development and evolution [33]. For example, Lee and coworkers [86] have reported that depletion of the CTCF protein induces selective cell death of cancer cells via p53. CTCF is specifically involved in DNA spatial reconfiguration and in the formation of the so-called topological associated domains.

Histone deacetylases (HDACs), DNA-methyltransferases (DNMTs) and chromatin remodelers [87], also exert action upon the gene regulatory program by adding or removing functional chemical groups to chromatin, thus altering the spatial configuration of DNA. These dynamical processes have well-documented roles in cancer phenotypes, HDACs have been reported to participate in malignancy [88,89,90,91] and to act in concert with miR regulators [92]. HDACs have actually turned into promising epigenetic therapeutic targets [89], their inhibition has been able to potentiate immunotherapy effects on triple-negative breast tumors [90], and to slow down tumor growth when mTORC1 and α-estrogen are also inhibited [88]. Regarding DNMTs, their activity throughout tumorigenesis and their potential as cancer biomarkers has been discussed extensively by Zhang et al. ([93]).

## 3. Omic Developments to Study Cancer Genomics

During the twentieth century several features of cancer, including cell-cycle checkpoints deregulation, oncogenes, tumor suppressor genes, genetic instability, and cancer gene interactions were identified and studied with cytogenetic and classical genetic techniques [94]. As a result of this knowledge, new questions surfaced about the regulatory context of genes that are significant to cancer processes, and two things that have been crucial to advance cancer research in the last twenty years into an integrative field, the availability of the human genome sequence [95] and the development of high-throughput sequencing methods. Today it is recognized that cancer genomes, transcriptomes, and epigenomes are all key to understanding cancer biology in a detailed molecular level, which has led to several combinations of different experimental methods with high-throughput sequencing to process cancer samples and generate quantitative data of the molecules that result from the genomic mechanisms behind malignancy. In this section, we describe genomic and epigenomic methods that are currently used to investigate the cancer phenotype. Throughout, we discuss the goals of the methodologies, their general workflow, data analysis, and challenges.

### 3.1. Sequence-Based Methods

#### 3.1.1. DNA Sequencing

DNA sequencing is used to determine the presence of potentially pathogenic (e.g., loss of heterozygosity) genetic variations (see [43]), including single nucleotide variations (SNVs), insertions and deletions (INDELs), copy number variations (CNVs) and genomic rearrangements. The aim of a particular DNA sequencing assay can be either detecting inherited germinal variation or characterizing somatic variation in tumor samples. In the former case, DNA is usually extracted from buccal swabs or blood samples. In the latter, it must be extracted from tumor cells, which are habitually derived from paraffin-embedded biopsy samples [96] or fresh-frozen tumor tissue [97]. However, there have been efforts lately to achieve high-quality DNA sequencing from minimally-invasive liquid biopsies [98].

##### Experimental Strategies

Although whole-genome sequencing has the advantage of uncovering the whole set of variations in a given tumor, sequencing a subset of the genome such as whole-exome or targeted regions is sometimes favored due to the lower cost and input DNA requirements, particularly in clinical settings [99]. Experimental design can accommodate a broad range of objectives, from uncovering key driver mutations and their frequency in large groups of individuals presenting a specific tumor type and identifying pan-cancer significant variation [100], to detecting germline oncogenic variants or guiding therapeutic options based on surveying tumor somatic variation in an individual [101].

After DNA extraction and quantification, library preparation can follow different strategies [102], for whole-genome sequencing the DNA is fragmented and sequencing adapters are ligated to it. However, when the goal is to sequence the exome or selected regions it can be done through ‘capture’ where a pool of diverse oligonucleotides are used to hybridize the genetic material of interest [103] or through ‘amplicons’ that are primers designed to flank the target regions and amplify them through Polymerase Chain Reaction (PCR) [104]. Although the obtained data from the different platforms are equivalent (see [105] for a comprehensive review), Illumina (see sub-section ’High-Throughput DNA Sequencing’ below) is currently the most used DNA sequencing technology.

##### Variant Discovery Analysis

Once the sequencing reads are properly mapped there are two procedures that correct for technical biases and should be applied to the data before carrying out variant calling. The first one is Indel Realignment and it is a de novo assembly of reads from regions detected to probably contain an insertion or deletion, but are mistaken for different SNVs close to each other, indel realignment algorithms include ABRA [106] and IndelRealigner from the Genome Analysis Toolkit (GATK) [107]. The second procedure is Base Quality Score Recalibration (available as a GATK tool) and it uses machine learning to analyze the sequencing read’s Phred scores and corrects their values accounting for systematic errors from the sequencer machine, this avoids over- and under-estimations from the subsequently used variant caller algorithm.

##### Germinal Variants

Variant calling is the identification of loci in the genome where the data that is being analyzed presents differences compared to the reference genome. One of the most used variant callers for SNVs and INDELs is HaplotypeCaller [108] from GATK, which evaluates individually each site of potential variation using as expected model a De Bruijn graph [109] built from the sequence of the reference genome and comparing the sequenced reads against it to obtain the list of observed possible haplotypes and variants, the best-supported genotypes throughout the samples are obtained through a Bayesian approach. Other algorithms include MAQ [110], Freebayes [111], which is useful when analysing amplicon sequencing data, BIC-seq [112] and FermiKit [113], that can additionally handle CNVs detection.

These algorithms output files in the Variant Call Format (VCF) [114] with ‘raw’ variants, that should be subject to filtering and annotation (see variant filtering and annotation). Germline variants are usually investigated to assess the risk of inherited cancer susceptibility and identify affected pathways [115], however it should be noted that the frequency of pathogenic variants can be different throughout populations [116,117] and also that most cancers are ‘sporadic’ as opposed to familial [118].

##### High-Throughput DNA Sequencing

**Illumina sequencing by synthesis** Illumina sequencing requires that adaptors are added through PCR to both ends of the DNA fragments, oligonucleotides complementary to the flow cell where sequencing takes place are also added. Additionally, indices may be included too when different samples are sequenced on the same run. The flow cells are made of lanes that are coated in oligonucleotides that hybridize and physically attach the DNA to be sequenced. Once the DNA library is loaded into the flow cell, each bound fragment is amplified clonally, generating clusters of around one thousand copies of each single-stranded DNA molecule, which improves the signal. Then, all the fragments are extended in parallel by polymerase enzymes one nucleotide at a time using nucleotides (dNTPs) with an attached fluorophore which also serves as a reversible terminator. In each extension cycle, the four dNTPs are added separately and the flow cell is imaged to identify the base that was incorporated through the fluorophore emission wavelength and intensity. The base calls are made in real-time from the images by the machine’s internal software. This extension cycle is repeated n times, resulting in sequencing reads of n bases. Currently, Illumina sequencers output a maximum read length of 300 base pairs.

**Sequencing data processing** The first step in the analysis of sequencing data in assessing the quality of the raw reads, which are usually in FASTQ format [119], this task can be performed with readily available tools such as FASTQC [120] or MultiQC [121] that summarize the data attributes with descriptive statistics of Phred quality [122], nucleotide content and sequence length distribution, among others. If required, adapter sequences are removed and reads are trimmed to eliminate low quality and ambiguous bases, in this case too, software like Trimmomatic [123] or Cutadapt [124] can be used. The next step is the alignment of the sequencing reads to a reference genome and there is a vast offer of tools that accomplish this job [125] and output a sequence alignment map (SAM/BAM) file [126]. Afterwards, some standard data cleanup is done, including ordering the mapped reads by chromosome and position and marking the reads that are determined to be PCR duplicates by their identical start and end positions. This last step is omitted when the used library strategy is amplicon sequencing, which by definition have very similar genomic positions.

##### Somatic Variants

Calling somatic variants can be challenging due to a number of factors that include, heterogeneity in tumor samples [127], the high diversity among driver mutations even between cases of the same cancer type, and the fact that most genetic changes present in cancer are not the drivers of malignancy. Additionally, a high depth of sequence coverage is usually required to detect somatic mutations accurately. Several somatic variant callers with diverse algorithm strategies exist nowadays, Bohannan and Mitrofanova provide a review of these tools in the context of experimental cancer biology [128], while Zare et al. evaluate the detection performance of variant callers [129].

The analysis of somatic variations in whole-genome and whole-exome data has led to the identification of recurrent and significant pathogenic alterations, including SNVs, INDELs and CNVs [130,131], which has helped elucidate patterns of genomic anomalies in cancer phenotypes. Moreover, the work of consortia like The Cancer Genome Atlas (TCGA) [132], now integrated with Genomic Data Commons (GDC) [133], found in different cancer types the most common driver mutations, as well as genomic profiles associated with prognosis, including hypermutation, microsatellite instability, content of CNVs, mutational burden, inactivating mutations in chromatin modifiers, DNA repair pathways and immune system genes [134]. A great advantage of efforts like GDC is that the data generated by them has great quality and is publicly available to the scientific community, which can enhance discovery through reanalysis [135].

##### Variant Filtering and Annotation

Raw variants are filtered to remove false-positive calls. A set of criteria including alignment quality, depth, read support of the reference versus alternative alleles and strand bias is used to calculate the probability that each variant call is correct. There are also machine learning methods, such as VQSR from GATK, that build estimators using data sets of previously known true variants, however, they require thousands of raw variant calls to operate on. Importantly, SNVs and INDELs are filtered separately because their properties are different. Afterwards, variants are annotated using genetic variation databases like 1000 genomes, gnomAD, or using databases of functional categories and predicted effects just as snpEff [136], dbNSFP [137] or ClinVar [138]. Furthermore, putative predictions about variants can be made according to proximity to regulatory regions, coding regions, and splicing sites, among others.

#### 3.1.2. RNA Sequencing

The introduction of microarrays to analyze gene expression [139] of thousands of genes simultaneously in an unbiased fashion unlocked the development of discovery-driven research and with the advent of RNA-sequencing [140,141] it emerged that studying the molecular phenotype of cells through the quantification of their whole-genome transcriptome is a very powerful tool to approximate the functional state of cells and large projects have been established to characterize not only transcriptomes from diseased phenotypes [142,143], but also from healthy ones [144].

**RNA-seq experiment** RNA-seq methods are based on the conversion of extracted and fragmented RNA to complementary DNA (cDNA) by a reverse transcriptase using random primers. The obtained cDNA, which is double-stranded, can be used then to build a sequencing library. It is common practice to take steps that enrich for messenger RNA “poly(A) enrichment” or deplete ribosomal RNA, due to the overwhelming abundance of the latter in cells and usual interest in the coding transcriptome. Actually, the number of RNA-seq protocol variations nowadays is staggering (recently reviewed in [145]) because they are suited to a variety of specific research goals.

**RNA-seq analysis** The main principle of RNA-seq analysis is that the number of sequencing reads for a given transcript, usually explicitly ligated to a gene, is a proportional measure of its expression level. RNA-seq is benefited from paired-end reads and depending on the objective of the assay a depth from 50 million reads, in the case of differential gene expression analysis, to 200 million reads for de novo transcriptome assembly.

The alignment of reads is usually to a reference transcriptome, however in RNA-seq from tumor samples, this step is sometimes done through de novo assembly because the alterations that are common in cancer give rise to transcriptional differences compared to the regular genome. Reads that map to multiple locations are usually filtered out since it is extremely difficult to distinguish their origin.

Assigning reads to genes or transcripts and quantifying them [146] represents a critical step for the final results and ideally should consider transcript variants, however, this is not always possible because all splice junctions are not necessarily sequenced.

Once the transcript counts are obtained, it is necessary to normalize the data to correct for biases that arise from reading depth, GC content, and intrinsic noisy differences between samples. Normalization methods have been developed since the introduction of RNA-seq and selecting one should consider the experiment attributes and goals [147], however, it is common practice to apply different normalizations to the data and compare the results to ensure the best possible results.

Finally, one of the most common goals of RNA-seq is differential gene expression (DGE) analysis, which seeks to determine the genes that are over and under-expressed in a condition compared to others. Available DGE methods and tools have been subject to evaluation [148,149] in the interest of assessing their fidelity and replicability, as well as guiding analysis decisions. Other RNA-seq analysis applications, provided the experimental design is suitable, include detecting transcript abundance from alternative splicing events, detecting gene fusions and evaluating the expression of transcripts that contain SNVs, among others.

### 3.2. Epigenomic Modifications

#### 3.2.1. DNA Methylation

DNA methylation is an epigenetic alteration with a role in transcription regulation, gene silencing, and chromatin organization [150]. DNA methyltransferases (DNMTs) add a methyl group on the fifth carbon of the cytosine ring, 5mC, avoiding TF binding. Instead, DNA methylation creates a binding site for Methyl-CpG-binding domain (MBD) proteins, which in turn recruit histone-modifying complexes [36]. Bisulfite causes the differential deamination of cytosine and 5 mC. While cytosine deamination turns the base into uracil, 5mC deamination is still detected as cytosine [151]. Then, bisulfite treatment followed by sequencing or microarray reading can effectively identify the modification.

Illumina Infinium arrays are the most common detection method [152]. The most recent one, MethylationEPIC BeadChip, can interrogate over 850,000 methylation sites across the genome. Methylation BeadChips measure trough fluorescent dyes the intensity of the methylated and unmethylated signal. Afterward, data has to go through quality control, background correction, and normalization. Illumina’s GenomeStudio software cope with the whole preprocessing but can be customized through Bioconductor packages such as IMA [153], Minfi [152], and MethyLumi [154]. Biologically variant regions like SNPs and sex chromosomes should also be filtered [155].

Though GenomeStudio provides an internal control normalization, results suggest that peak-based correction and between arrays quantile normalization plus Beta-mixture quantile normalization within arrays may outperform others [156]. The beta values outputted by this pipeline must be transformed to M values and corrected for batches, but are otherwise ready to be used. beta values give account for hypo and hypermethylated regions, hence, a bi-modal distribution is obtained. M-values transformation normalize bi-modal distribution, in order to perform further analyses.

#### 3.2.2. Chromatin Immunoprecipitation Followed by Sequencing (ChIP-seq)

Normal transcriptional programs in healthy cells are largely controlled by TFs encoded in the human genome [157], that bind specific regulatory sequences in the genome and activate transcription of their associated genes by recruiting cofactors, chromatin regulators, and the RNA polymerase II (RNAPII). Moreover, the main component of chromatin are nucleosomes that consist of histone proteins that can be modified post-translationally by covalent addition of a functional group to amino acid residues in their C- and N- terminal domains, which has a role in transcriptional control (reviewed in [158]).

Therefore, the patterns of TF binding and chromatin modifications have been investigated for their role in gene expression changes and clinical outcomes in cancer [159,160,161,162,163,164]. Chromatin immunoprecipitation followed by sequencing (ChIP-seq) [165,166] is a genome-wide method that detects in vivo binding events of TFs, positions of histone modifications or genomic presence of chromatin-associated non-histone proteins.

A validated highly specific antibody to the protein or modification of interest is necessary and depending on the abundance and stability of the target, from 10^4^ to 10^7^ cells. Frozen tissue may be subject to ChIP-seq, however, attention must be given to the homogeneity of the sample, because it can affect the target signal. The Encyclopedia of DNA Elements (ENCODE) consortium has developed guidelines and best practices for ChIP-seq assays [167] that can be useful in planning experiments.

**ChIP-seq experiment** Cells are crosslinked using formaldehyde [168], which fixates DNA and proteins in vivo, the chromatin is sheared through sonication (although protocols using enzymatic digestion are available) that is optimized to generate fragments ranging from 100–600 bp in length and the antibody is used to immunoprecipitate the DNA fragments bound to the target of interest. Finally, after crosslink reversal to allow for DNA purification, a sequencing library is prepared and sequencing is used to profile the events of interest across the genome. There are three types of controls for a ChIP-seq experiment, (i) input DNA, which is taken from the sample before immunoprecipitation (IP), (ii) mock IP DNA, from IP without using an antibody, and (iii) nonspecific IP DNA, from IP with an antibody against a non related protein, commonly IgG, additionally, to account for inter-sample variation a replicate experiment is recommended. Single-end sequencing is often used in ChIP-seq experiments, with the benefit of paired-end sequencing being increased mappability, while the optimal sequencing depth is in the function of the type of the experiment’s target and is around 30 million and 60 million mapped reads for DNA-binding proteins and histone marks, respectively [167].

**ChIP-seq data analysis** Once the raw sequencing reads have been aligned to the reference genome and appropriately filtered, peak calling is performed to identify genomic locations enriched for the targeted protein. The goal of peak calling is to find regions flanked by reads on both ends (5’ and 3’), deemed candidate peaks, and evaluate them statistically versus a background model (reads from the control experiment or expected values in the matching region) to assess their significance.

Some of the most used peak callers (for a comprehensive review see [169]) are MACS [170,171], SPP [172], and PeakSeq;
the first two use a Poisson distribution to model the data and calculate the cutoff above which a peak is determined significant, while SPP uses a binomial distribution and considers the mappability of the regions.

It is important to stress out that peak calling is sensitive to user-defined threshold values. The output of the analysis is a list of the genomic regions where significant evidence of binding/presence exists, along with *p*-values and false discovery rates (FDRs). Downstream analyses include protein binding motif discovery, differentially enriched regions analysis and integration with RNA-seq gene expression data.

#### 3.2.3. Methods to Assess Open Chromatin

Regions where nucleosomes are sparse and physical access to the DNA sequence is enabled are identified as open chromatin. Chromatin accessibility is a dynamical and complex framework modulated by diverse elements, including nucleosome occupancy and turnover rate, histone modifications, ATP-dependent chromatin remodeling complexes, and even TF binding [26,173]. Open chromatin has emerged as indicative of transcriptional regulatory potential or activity across the human genome because most of the TFs analyzed to date bind within open regions [28].

Particularly in the context of cancer, a large survey by the TCGA analysis network [29] revealed a general increase of open regions in cancer compared to healthy phenotype, a connection between susceptibility genetic variants and accessible chromatin, as well as grouping of breast cancer and kidney renal carcinoma samples by the presence of open chromatin peaks, that turned out to be accompanied by gene overexpression and clinical implications. Moreover, studies that interrogate chromatin accessibility have helped to uncover specific TFs that play a role in the gene expression patterns of tumor samples [174]. Hence, assaying open chromatin can help researchers gain knowledge of the processes deregulated in the transition of normal cells to cancer.

Several methods exist to assess open chromatin sites, DNaseI-hypersensitive (DHS) sites derived from DNase-seq [175] coincide with nucleosome-depleted regions and Micrococcal Nuclease sequencing (MNase-seq) [176,177] experiments help determine positions where nucleosomes are present, while both techniques rely on enzymes to digest unbound, open chromatin, formaldehyde-assisted Isolation of regulatory elements sequencing (FAIRE-seq) [178] takes advantage of the different chemical properties between protein-bound DNA and nucleosome-depleted DNA. However, since the introduction of the assay for transposase-accessible chromatin using sequencing (ATAC-seq) [179], most investigations favor it, due to its simplicity and low DNA-input requirements, and there is even a follow-up method that couples ATAC-seq with high-resolution microscopy [180]. It should be noted that the data produced by these methods present a high correlation, and it has been proposed that the differences arise mostly from sequencing biases [181].

#### 3.2.4. Transposase-Accesible Chromatin Sequencing

**ATAC-Seq experiment** ATAC-seq is a clever method that leverages the activity of a transposase called Tn5 to simultaneously fragment and tag all the accessible DNA. The enzyme is pre-loaded with sequencing adapters, allowing for direct purification and amplification of the fragmented and tagged DNA, followed by sequencing. Usually, two replicates are used to discern biological signals from noise and while appropriate sequencing depth can depend on the target cells, the original protocol [179] suggested around 50 million aligned reads per sample.

**ATAC-Seq data analysis** After the alignment to the reference genome and corresponding filters, peak calling is carried out to identify statistically significant enrichment of reads throughout the genome. Essentially the same peak callers employed in ChIP-seq analysis can be used for this type of data. Importantly, a blacklist of ATAC-seq peaks from ENCODE [182] is available to filter the results. When multiple samples are available a custom strategy to obtain high-confidence open chromatin peaks can be developed, for example, using criteria of presence in more than *n* samples or normalizing peak significance score and using a threshold thereafter. With the ultimate goal of characterizing and better understanding which regulatory landscapes may underlie the studied phenotypes, downstream analyses to an ATAC-seq peak set include annotating them with data from external sources [183] to find coinciding histone marks and/or DNA-binding proteins, searching for enrichment of TFs binding motifs [184] or footprinting analysis to derive a measure of TF occupancy [185,186].

### 3.3. Chromosome Conformation Capture (3C Methods)

In 2002 Dekker et al. introduced an innovative technique called 3C [187] to measure at high resolution the frequency at which any two genomic loci, for example, enhancer and promoter, were found together in the nuclear space. This opened exciting avenues in the investigation of the three-dimensional conformation of the eukaryotic genome, whose structured nature had been recognized [188], but was almost exclusively studied with microscopy methods [189]. The 3C technique was followed by the development of assays to quantify chromatin interactions between all the loci within a defined region at the Megabase scale (“5C” [190]), between a viewpoint and the rest of the genome (“4C” [191]), and the genome-wide interactions (“Hi-C” [192]).

Soon thereafter, general patterns of the conformation and interactions within the chromatin framework emerged, including transcriptionally-repressed lamina-associated domains [193,194], A/B compartments that roughly correspond to euchromatin and heterochromatin [195], topologically associating domains (TADs [196]) that interact mostly within themselves and chromatin loops between regulatory sequences [197] formed by CTCF sites in convergent orientation. Proteins involved in the architecture of the 3D chromatin structure were also identified [198] and today it’s well accepted that genome organization is linked to a myriad of functional processes, such as developmental regulation, gene expression or silencing throughout the cell cycle, DNA repair and deregulation in disease phenotypes.

The role of the 3D organization of the genome in genetic regulation is an ongoing and quite active research field, it has spawned variations of the C methods that are tailored to regulatory genomics questions, for example, chromatin interaction analysis by paired-end tag sequencing (ChIA-PET [199]) to detect chromatin interactions mediated by a specific TF or protein, capture Hi-C (CHi-C [200]) to identify interactions between specific regions of interest and the rest of the genome, Hi-C methods to achieve kilobase resolution [201,202], to obtain contact maps from clinically available samples [203] and even to unmask the processes behind chromosome interactions through the quantification of their stability [204]. In spite of the buoyant progress in the research of chromatin’s functional structure, the characterization of its direct relationship to transcriptional regulation is work in progress [205,206,207,208].

Through the application of the C methods it was identified that the three-dimensional architecture of chromatin is correlated to the presence of somatic alterations in cancer [209,210,211], and even though Hi-C measures interaction frequency and not physical distance [212], the former can be a location predictor of chromosomal rearrangements and CNVs in cancer [213,214]. These alterations of the DNA sequence, which are typical in tumors, can lead in turn to disruption of the chromatin framework in which regulatory interactions take place [215], resulting in oncogene activation due to aberrant contacts between a foreign enhancer and their promoter [216,217,218].

In light of this, there have been efforts through C methods to identify non-coding alterations that impact gene expression and drive cancer progression [219] and to profile the regulatory loops that impact transcriptional programs in a clinical research context [220]. Indeed, when ChIA-PET was used to investigate the relationship between TFs mediated by hormones, namely the estrogen-receptor-alpha (ER-alpha), chromatin interactions and the transcriptome in the context of breast cancer, it was suggested that the coordinated regulation of sets of genes could be aided by their co-localization in space mediated by the RNA Polymerase II and that the perturbation of this arrangement can lead to transcriptional alterations of even secondary genes [199,221]. Later, two studies [34,222] reported that upon activation of the ER-alpha, transcriptional changes entail coordinated responses at the chromatin structure level. Finally, Hi-C experiments of genome-wide chromatin interactions have identified a switch from B to A compartments accompanied by up-regulation of their resident genes in breast cancer [223] and B-cell lymphoma [224].

#### 3.3.1. Genome-Wide Chromosome Conformation Capture

**Hi-C experiment** The Hi-C assay begins with the fixation of the DNA using formaldehyde to preserve the cellular conformation of the chromatin. Afterward, a restriction enzyme that leaves sticky ends is used to digest the DNA, it is important to note that the resolution of the data will depend on the frequency of the enzyme’s restriction sites in the genome. The overhanging ends are filled with biotinylated nucleotides and religation is promoted between DNA molecules that belong to the same interaction complex; this step can be performed under dilute conditions [192] or in the cell nucleus (in situ Hi-C [201]). Finally, the crosslink is reversed, followed by sonication and biotin pull-down using streptavidin to enrich for the chimeric DNA of interest, which is amplified and sequenced.

**Hi-C analysis** There are two excellent reviews [197,225] that provide guidelines to analyze data from Hi-C experiments. Although there are several options at each step of the analysis, we describe one of the most common workflows. Briefly, the sequencing reads should be cleaned, specifically, they should be scanned for ligation junctions (two restriction sites facing each other) and trimmed to improve mappability. The paired-end reads should be aligned to the reference genome separately, because the pairs do not correspond to contiguous sequences in the genome; a widely used strategy is iterative mapping in which all the reads are trimmed to n nucleotides and aligned to the reference, the reads that do not align are extended by n nucleotides in the 5’ direction and mapped again. Aligned read pairs are assigned to the nearest restriction fragment in the genome and filtered to retain only informative pairs. Afterward, the genome is binned in fixed-length windows and the read pairs are assigned to them which yields a contact matrix that must be normalized before meaningful interactions can be derived. Several methods to normalize Hi-C matrices exist, explicit-factor correction [226] considers known biases as mappability and GC content to calculate the probability of contact, while matrix balancing [227] is an implicit correction method based on the Sinkhorn–Knopp balancing algorithm [228] that results in a Hi-C normalized matrix with all the rows adding up to the same quantity. It is important to note that matrix balancing requires that the raw matrix is filtered to mask bins with very few read pairs. Once a Hi-C matrix is normalized, there are different methods to obtain compartments, TADs and significant contacts from it, they are richly discussed in [197]. Afterward, the data can be integrated with data from location-based methods, including ChIP-seq and ATAC-seq, to profile sets of interactions that are considered of interest.

#### 3.3.2. ChIA-PET

The Chromatin Immunoprecipitation Analysis by Paired-End Tag Sequencing combines formaldehyde-crosslinking to obtain chimeric DNA molecules that are in nuclear proximity and enrichment for a subset of loci by means of a specific antibody to probe regulatory chromatin interactions mediated by a protein of interest. It should be noted that while ChIA-PET can provide de novo, unbiased short- and long-range chromatin interaction profiles, the protein involved must be previously suspected to participate in mediating these functional contacts. Usually, the assayed protein is a TF, a form of the RNA polymerase (i.e. initiation or elongation RNAPII) or a structural protein (e.g., CTCF). Some applications of ChIA-PET include identification of chromatin contacts by a TF between promoters and other regulatory sequences, evaluation of differential interactions in a myriad of phenotypes (e.g., developmental stages, response to cell signaling, disease processes), and notably, characterizing the spatial nature of miRNA genes transcriptional regulation [229].

**ChIA-PET experiment** The ChIA-PET experiment, as described in the original protocol [230], begins by crosslinking the DNA in the nucleus followed by sonication to lyse the cells and release the fragmented chromatin. A highly-specific and validated antibody is used to immunoprecipitate genetic material bound by the protein of interest, this increases the specificity of the library and reduces background noise. As in other IP-based protocols, a sufficient number of starting cells is required (the original protocol estimates 100 million cells) to achieve adequate library complexity. After the DNA-protein complexes are immunoprecipitated, biotinylated oligonucleotide half-linkers that contain a recognition site for the MmeI restriction enzyme are ligated to the free ends of the chromatin fragments. Two half-linker variants (A and B) with specific nucleotide barcodes to distinguish them are used; prior to ligation the ChIP chromatin is divided into two aliquots and each is ligated with half-linker A or B. Both fractions are integrated under dilute conditions to promote DNA inter-ligation, thus creating three types of junctions (heterodimer AB linkers and homodimer AA/BB linkers) that help distinguish non-specific ligation products. The crosslinking is reversed, the DNA is purified and the type IIS MmeI enzyme is used to release the tag-linker-tag constructs that are captured using streptavidin and finally used to build the sequencing library.

**ChIA-PET data analysis** The analysis of ChIA-PET data is complex and different methods have been proposed to extract meaningful contacts from the million reads usually obtained by an experiment, ChIA-PET tool [231] solves the problem by means of a hypergeometric model, while model-based Interaction calling from ChIA-PET data (MICC [232]) employs a hierarchical mixed probability model and ChIAPoP [233] relies on a Poisson model. However, the first step in the analysis involves classifying the read pairs in heterodimer and homodimer linkers and aligning the tags to the reference genome plus the usual post-mapping filters. Then, the overlapping read pairs are merged and peak-calling is performed to locate enrichment loci within the genome, this results in “paired” peaks, since they are located at different positions in the genome. The paired peaks correspond to pairs of loci connected by the immunoprecipitated protein, although it should be noted that ChIA-PET cannot determine if the assayed protein is directly responsible for the interactions or if it simply present e.g., as part of a protein complex.

### 3.4. Genome and Epigenome Editing

Studying in vivo cancer biology is a daunting task complicated by the fact that the processes governing the cancerous abnormal regulation of the genome are not yet fully characterized. For example, the onset and initial steps of tumorigenesis are hardly ever observed or probed, the contribution of most of the detected variants to different cancer types is not clear, and the tumor microenvironment cues influence dynamically the abnormal regulatory processes [234]. Therefore, even when formidable advances in cancer research have been accomplished through the application of the high throughput methods outlined above, perturbation experimental approaches are required to dissect the genotypes underlying most cellular and molecular phenotypes of carcinogenesis.

Perturbation screens [235] act at the DNA, RNA or protein levels to gain insight about gene cellular functions and essentiality, but also to better understand the intricate regulatory mechanisms of the genome and find drivers of disease [236] as opposed to its secondary manifestations. While earlier strategies like retroviral insertion/transposon mutagenesis produced random perturbations, nowadays sequence-specific, genome-wide methods are available to perform DNA sequence [237], transcriptional [238,239] or post transcriptional [240,241] perturbation screens. Directed assays also enable multiplexed or pooled screens [242] and a higher degree of precision. Particularly, the development of clustered regularly interspaced short palindromic repeats (CRISPR) editing [243] revolutionized the entire functional genomics field and has become an obvious choice [244] to perturb gene activity due to its minimal interference with endogenous conditions, versatility and relative simplicity.

#### 3.4.1. Clustered Regularly Interspaced Short Palindromic Repeats (CRISPR)-Cas

CRISPR is the only known instance of acquired immunity in prokaryotes. It consists of a CRISPR locus in the bacterial genome that is actually foreign in its origin and usually corresponds to the genetic sequence fragments of a bacteriophage or plasmid. When a bacteria has acquired CRISPR DNA [245], it can be transcribed and processed into a mature crRNA that is used by CRISPR associated (Cas) proteins to detect and cleave pathogenic DNA from similar viruses upon reinfection. The Cas proteins are DNA endonucleases that undergo a conformational change that allows cleaving as a consequence of the base-pairing (around 20 nucleotides) between the crRNA and the foreign DNA. Additionally, The action of CRISPR-Cas depends on the presence of a protospacer adjacent motif (PAM) in the target DNA that has been theorized as a bacterial non-self recognition system.

**CRISPR-Cas genome editing in cancer research** Several research groups identified and characterized elements of the CRISPR-Cas system [246]. However, when a team led by Jennifer Doudna and Emmanuelle Charpentier engineered guide RNA to direct Cas9 cleavage in a sequence-specific manner [243], they rendered a programmable genome-editing tool unparalleled in power to silence or activate specific genes. Although the initial employment of the Cas nucleases was inducing double-strand breaks in specific loci that result in inactivating frameshifts, currently numerous alternatives have been developed to modulate gene expression [247]. Briefly, the nuclease domains from Cas proteins can be deleted while preserving targeting function through the crRNA. This can be exploited through protein fusions with other enzymes that are transcriptional activators or repressors, epigenetic effectors that modify chromatin and even enzymes to induce point mutations [248]. Additionally, imaging techniques that couple catalytically inactive Cas9 to fluorescent proteins have also been developed with potential applications in tracking endogenous RNA activities and post-transcriptional regulation [249].

CRISPR-Cas has generated promising strategies in translational cancer research [250,251]: from uncovering relevant mutations to establishing exemplary animal models and carrying out pooled perturbation assays (see the ’CRISPR Screening Assays’ sub-section below), but it has also put forward impressive therapeutic approaches like correcting faulty DNA sequences or enhancing immune system cells to fight tumors. Importantly, methods that use CRISPR interference to probe the target promoters of enhancers [252] could help clarify the participation of non-coding mutations in cancerous gene regulatory processes, albeit different mechanisms of enhancer activation should be taken into account [253].

##### 3.4.2. CRISPR Screening Assays

**Pooled screens** The goal of a pooled CRISPR screen is to assess the effects of genome-wide perturbations simultaneously in a single assay (in contrast to array screens where perturbations are assessed individually). Often, the idea is to establish a causal relationship between a genotype and phenotypes of interest or to test the functions of genes under different contexts. For example, a pooled screen can be used to evaluate phenotypes like cell death, drug resistance or transcriptome changes.

Broadly, a CRISPR screen assay is performed as follows. However, it should be noted that considerable effort needs to be put into experimental design and subsequent experiments planning in order to obtain reasonable results [244].
A library of perturbations is built according to criteria largely determined by the assay’s goal.The library is introduced into a population of cells using a vector that could be a lentivirus or retrovirus.The cell population is screened to select the subset of cells that present the phenotype of interest.PCR is usually employed to amplify the sequences from the genomic DNA and identify which perturbations gave rise to the phenotype.Finally, the relative abundance of the perturbation library can be quantified by DNA sequencing.

**Technical limitations** Standardizing the perturbation library introduction is important to obtain reproducible results. Additionally, the delivery system should be thought out carefully, especially in the in vivo models to avoid potential immune responses against CRISPR [254] or plain delivery failure. Finally, a highly relevant technical limitation is the off-target activity of Cas proteins that can edit wrong loci, notwithstanding CRISPR-Cas’ unmatched precision [255]. It has been proposed to leverage different PAM sequences and their specificities to alleviate this problem [256,257] and a sequencing-based method has been developed to detect genome-wide double-stranded breaks by CRISPR RNA-guided nucleases [258]. More recently, anti-CRISPR molecules derived from phages have been used to block CRISPR-cas activity in mammalian cells [259], however, the authors had in mind the development of controllable synthetic gene circuits rather than CRISPR pooled assays.

## 4. The Need for an Integrating Framework

In the previous section, we have presented and discussed some of the most representative omic experimental approaches to characterize different facets of gene regulation patterns in cancer, as well as their main computational analysis approaches and bioinformatic tools [260]. Since each of these approaches contributes in different ways to the global phenomenon, there is a need to find theoretical frameworks and methodological techniques to integrate the knowledge derived from each of these omic technologies into a coherent, hopefully, mechanistic explanation of gene expression deregulation in cancer [11]. As is mentioned, there is a wide diversity in the types of data, dynamic ranges, sources of noise and error and other features, a fact that further complicates the development of such a holistic, integrated approach. Some preliminary proposals have been outlined in recent times [5,12]. Such proposals often combine one or more of the approaches presented in Figure 2. In this section, we will present some general methodological concepts that have resulted in relatively successful ways to integrate different classes of omic data and ultimately inform about the complex underlying cancer gene regulation patterns. Multiomics integration borrows different techniques from statistics, while multivariate analysis tackles classification and regression problems involving distinct molecular levels, probabilistic methods link entities by their chance of occurring independently of their nature creating networks that can be further explored (although it should be noted that methods like Similar network fusion [261] rely on non-probabilistic networks). Finally, statistical learning allows for feature selection amidst the broadness of omics.

### 4.1. Computational Approaches to Omic Integration

Multiomics integration aims to harness the interaction between the different biological levels captured by the omics. It rests on the extraction of complementary information, measured ideally on the same set of samples, in order to find co-occurring patterns [262]. Achieving such a goal will enhance the study of biological phenomena by improving models, not anymore limited to the scope of a single kind of experiment.

#### 4.1.1. Omic Integration Stages

##### Late Multi-Omic Integration

The so-called late integration analyzes each omic separately and then combines the individual results ignoring inter-omics effects [263]. Though simplistic, this approach excels in classification and prediction. An example is the clustering of gynecological tumors made by Berger et al. They classified tumors in a clinically relevant way, based on previously chosen cancer biomarkers ranging from gene mutation and CNV load to receptor protein expression and immune infiltration. By simply dichotomizing biomarkers into a matrix of samples per features, where zero indicated absent or low and one meant present or high, they flattened features to uniform units that can be clustered. This way, by diminishing differences between omics type of data and range, they manage to cluster major cancer subtypes from ovary, uterus, cervix, and breast in groups with significantly different survival that do not overlap with the histological classification [264]. Though this does not strictly qualify as multiomics integration since specific elements were given to the clustering algorithm, it demonstrates the richness hidden across molecular levels, which could even inform cross-tumor-type therapies. Since some cases of uterine cancer and Luminal A breast cancer with good survival share high CNV load, low immune infiltration, and high AR, PR and ER α protein expression, Luminal A therapies could work for uterine cancer cases with such characteristics. However, now that we know these features are linked, we have to ask how are these characteristics related.

##### Early Multi-Omic Integration

To answer the how sort of questions, an early integration that acknowledges interomic effects with no pre-established direction [263] is needed. However, integrating the measurement of the different molecules taken up by the omics is not easy. Each omic has its own output that does not have to fit in dimension, variance, scale or noise to any other. Fortunately, methods that deal with every one of these issues exist, or are being developed [265].

The forthcoming section tackles multivariate techniques for omics integration due to its capacity to model simultaneously different omics [266] and select data-relevant features automatically. Nonetheless, these are not the only techniques available. For alternatives and ready to use tools check Bersanelli and Huang reviews [267,268].

### 4.2. In the Beginning There Was Statistical Learning

Omics measure thousands of molecules in tens to hundreds of samples. As a consequence, we the omics users, are by default suffering from the curse of dimensionality. When the number of subjects, *n*, is lower than the number of probes measured, *p*, we are subsampling the possible combinations of values and can hardly cover the actual space of study [269]. But this is no new problem, statisticians have been struggling with *p* >*n* for longer than omics have been around. Then, the severe *p*
>>
*n* hit when integrating different omics is not as intractable as first sight may suggest. Statistical learning, the area of statistics dedicated to model and understand complex datasets -exactly what multiomics want-, deals with *p*
>>
*n* situations applying regularization and reduction of dimension.

Regularization shrinks the coefficients on a model towards zero. Such shrinkage comes from a penalty added to the model fitting function. When shrinkage forces some of the coefficient estimates to zero, variable selection is attained [269].

#### 4.2.1. Least Absolute Shrinkage and Selection Operator (LASSO) Methods

The regularization method least absolute shrinkage and selection operator (LASSO), accomplishes variable selection through scaling of the l1 norm of the vector of coefficients with a tuning parameter λ. As a consequence of the penalization, predictors with the highest correlation to the modeled response get selected [269]. Further, even though their value does not necessarily translate to original measurements [270], coefficients weigh the importance and direction of the predictor’s effect [271], allowing deeper focus on selected variables. These properties have proven useful to find gene expression direct regulators, phenotype-specific molecules and interactions among distinct molecular levels [35,272,273].

Sohn and coworkers applied a LASSO multivariate linear regression to model ovarian cancer gene expression based on methylation, miRNA expression, and CNA. Results suggest there is a disparate impact of the regulators on the grade of expression. While highly expressed genes tend to associate with CNA; variable expressed genes are better explained by methylation features. When checking top coefficients, methylation appears as the omic with the strongest effect, with CNA coming lower in the rank and miRNAs dispersed all over. Then, the LASSO integrative setting, not only models simultaneously the distinct data types but allow a ranking of the omic’s effect though its feature selection capacity. Additionally, the network linking the predictors selected by the integrative models shows better modularity and specific functional enrichment than the one derived from non-integrative models, supporting the need for integrative studies [273].

The problem with the LASSO is that it selects at most *n* variables before getting saturated [274], which makes necessary additional steps for the *p*
>>
*n* multiomics case.

#### 4.2.2. Dimension Reduction

Dimension reduction surpasses LASSO saturation by exploiting the matrix representation of omics to find *M* < *p* linear combinations of the original predictors. The use of these linear combinations instead of the original predictors effectively lowers dimension from *p* to *M* [269].

The definition of data subspaces exploits matrix factorization techniques like the ones described in [275]. Both co-inertia analysis, CIA, and sparse partial least squares (sPLS) maximize the covariance between eigenvectors, while Canonical Correlation Analysis, CCA, maximizes their correlation [276]. Multiple factor analysis (MFA) projects a multi-omics matrix that can include both numerical and discrete datasets into the principal components subspace [266]. Joint and individual variation explained (JIVE) decomposes each omics matrix into joint, individual and residual variation structures [277,278]. CCA and JIVE can become sparse trough penalization, at least dimension reduction has been combined with LASSO penalization for both regression and classification settings [279,280,281,282].

Trough sparse multi-block partial least squares, Li et al. found multilayer gene regulatory modules on ovarian cancer data. Starting from 799 microRNA and 15846 gene expression profiles, 31324 loci with CNVs and 14735 DNA methylation marks; they reduced the high dimensional dataset into modules with an average of 45 CNV loci, 42 methylation marks, 5 microRNAs and 44 genes. Inter-omic relations found had significant IPA p-values, demonstrating the power of the technique [281].

#### 4.2.3. Elastic Net Regularization

LASSO saturation is paired with breakage of correlated groups, where just one of the grouped predictors was selected while the rest were discarded. The elastic net regularization overcomes this via a strictly convex optimization determined by an α parameter [274]. The α parameter determines how similar is the model to LASSO (α = 0) or to ridge regression (α = 1), another regularization method that is unable to select variables but gives similar coefficients to correlated variables. Thus, the elastic net is expected to recover variables than the LASSO does not, including both true and false positives [283,284,285].

Theoretically, multivariate methods recover synergistic effects that traditional paired correlation studies can not, due to the simultaneous analysis of the distinct omics [273]. The capture of entire groups of correlated variables suggests that the elastic net could overmatch LASSO on this task. Though simulation Neto et al. show the elastic-net outperforms LASSO predictive power when the variables are highly correlated. Examples of use can be found at [285,286]. Both LASSO and elastic net were used to explore the link between SNPs, DNA methylation and gene expression in bladder cancer in [284]. Results suggest SNPs and DNA methylation regulate cis gene expression, but each penalization method identifies distinct genes that would be in this situation.

#### 4.2.4. Feature Selection under Heterogeneous Dynamic Ranges

Either regularization method has two more issues. Omics matrices need to be concatenated no matter their difference in scale and; the same penalization is applied to all the omics ignoring that each molecular level might affect the response differently. Even when these problems arise at separate time points, they exhibit the same question, we are still learning how to fit matrices of different size and range together.

To interpret co-occurrent measures of different omics a normalization that assures all the information is effectively taken into account is necessary. Such normalization can be as abrupt as the mentioned dichotomization made by Berger et al. or as broad as centering around zero with unit variance, which is the recommendation of most methods [266,282,287]. However, it is extremely important since it can shape the final results. Situations where the largest dataset dominates may require to scale each data type by its total variation to force them to contribute equally [277] or, to scale each omic by its first eigenvalue to rest heavier on the more informative omics [266]. In this sense, the decomposition method AJIVE, being insensitive to scale heterogeneity [278], has an advantage over multivariate methods.

Bringing data to the same range does not guarantee a balanced variable selection across omics. Applying the same shrinkage could shrink to zero all the coefficients of subtler effect omics. To solve this, Liu et al. tunned the extent of penalization per omic achieving an optimal shrinkage for each omic. The resulting model achieved better classification than a single penalty over simulations and cancer samples with gene expression and methylation data [286]. Weighted penalization had been used by Lee et al. to deal with low and unbalanced sample numbers for breast cancer subtypes prediction [35], making the approach an appealing modification.

#### 4.2.5. Modeling Related Issues

Multivariate techniques applied to multi-omics deliver models for the interaction between omics. Thus, they carry the same concerns on statistical power and overfitting that all models have. Such problems are largely driven by limited sample size. Even though many more datasets are available now, there is still a lack of co-measured omics. The constrained sample size is nevertheless exploited via k-fold cross-validation plus testing on unseen data.

However, the sample size is not the only factor driving model quality. Small effects require larger detection power, which can be tuned through weighted penalties [286] as explained in the previous section. Significance can not be measured intrinsically, but p-values can be assigned following permutation approaches [284,288]. The threat of overfitting is lessened by the sparsity of the models [271], but it is still necessary to assess prediction accuracy in samples not used for training. Such testing with independent data, tests model validity by measuring coherence with published results.

Even with all the drawbacks described, multivariate omics integration has the undeniable advantage of unbiased variable selection. Future work is expected to set the guidelines for the reproducible application of penalization models with omics data [265,289]. Special efforts need to be done on the explicit report of model fitting and evaluation processes. In this sense, Git tools could be helpful. The incorporation of network strategies is promising too, as explained in the next section.

### 4.3. Network-Based Methods

One almost paradigmatic way to comprehensively map and analyze system-level (genome-wide) interactions in contemporary biology is by using complex networks. The network view has been used so extensively to integrate information of high throughput experiments in biology that, for some people systems biology has become almost a synonym with network biology [290]. Gene expression regulation at the whole genome level in cancer has been extensively studied in the past [291,292,293,294].

Network analyses have been used in the past to integrate multiple omics experiments in relation to gene expression regulation in cancer [65,67,295] and other diseases [296,297]. Multi-omic networks have also been discussed in other instances of biomolecular regulation [298,299]. The mathematical foundations to integrate and analyze such multilayer networks are however being laid out [300,301,302], and strategies for their use in multiomics are currently also under development [303].

For instance, an approach called similarity network fusion was developed [261] to integrate gene expression, DNA methylation, and miR expression data coming from five different cancer data sets. The method was useful to better ascertain cancer subtypes predicting survival. Costa and co-workers [295], in turn, used a multinetwork consisting of correlations among differentially expressed and differentially methylated genes in head-and-neck squamous cell cancer (both HPV+ and HPV-) to identify a set of genes with methylation alteration patterns in their promoter. They observed co-expression modules leading to discover key regulatory elements.

Finding molecular signatures was also the strategy followed by Gibbs et al. [296] that aside from disease-specific findings, provide a platform-agnostic means to study the relation between gene and protein expression at a genome/proteome wise level. Integrative multi-omic studies relying on network-based methods have been able to even provide a theoretical framework to study, not only the interaction structure of the gene regulatory maps, but also approximations to the kinetics and dynamics of gene regulation by means of the so-called static signal flow, and dynamic signal flow analyses [299].

### 4.4. Hybrid Approaches

Multivariate, statistical learning and network-based methods are largely complementary and that, powerful as they are on their own merits; neither of them is able to capture the full complexity and the subtleties associated with the integrative multi-omic characterization of gene expression programs in cancer. Both areas are indeed growing very fast, with a number of new methods and improvement of existing methods constantly been added to the current literature. A good way to exploit the capabilities of both approaches is, well, integrating them.

## 5. Concluding Remarks

Cancer is a complex disease. The way in which its molecular (genomic) origins move forward to the cellular, tissue, and phenotypic level is most often mediated by changes in gene expression regulation programs. Here, we have discussed that there is a multitude of disparate factors contributing to the reprogramming of the gene regulatory mechanisms. Among these, we have analyzed the role of changes in the action of promoters and enhancers, variations in the chromatin structural disposition, DNA chemical modifications as well as transcript splicing, stability and transport kinetics, all of the factors that have been known for a while. Also relevant to modify gene expression patterns, are more recently discovered factors, such as large chromosomal aberrations, like chromothripsis; the larger-scale effect (at the chromosomal level) of DNA 3D structure and the action of regulatory non-coding RNAs. Just a few years ago, it was impossible to characterize these effects at the global, whole-genome level. However, recent developments in experimental omic techniques have allowed us to measure those effects. Some of the more commonly used techniques were also summarized in this review.

Acknowledging that the genomic landscape is only a part of the whole picture, genome regulators have acquired more attention. Given that methylation profiles, ncRNAs, or 3D DNA structure have a strong influence on gene expression, to develop computational techniques that accurately measure small variations in different expression levels provide us tools that allow us to identify how these non-genomic variations directly affect the cancer genome.

Letting aside the computational challenges inherent to the analysis of single-omic experiments, these techniques however, have unveiled an enormous problem for computational oncogenomics: i.e., how to build models and integrate this enormous wealth of disparate information, into coherent and predictive models that will help us to decipher how the different faces of gene regulation interact to develop the anomalous gene regulatory programs that we deem responsible for the rise, establishment, and maintenance of the tumor phenotype; the well-known hallmarks of cancer. We discussed how the combination of powerful computational models based on statistical learning, machine intelligence, and probabilistic modeling, but also in network (and multinetwork) based methods, presents as an appealing alternative to be developed by the coming generation of computational cancer scientists to tackle the enigma of gene regulation in cancer.

## Figures and Tables

**Figure 1 genes-10-00865-f001:**
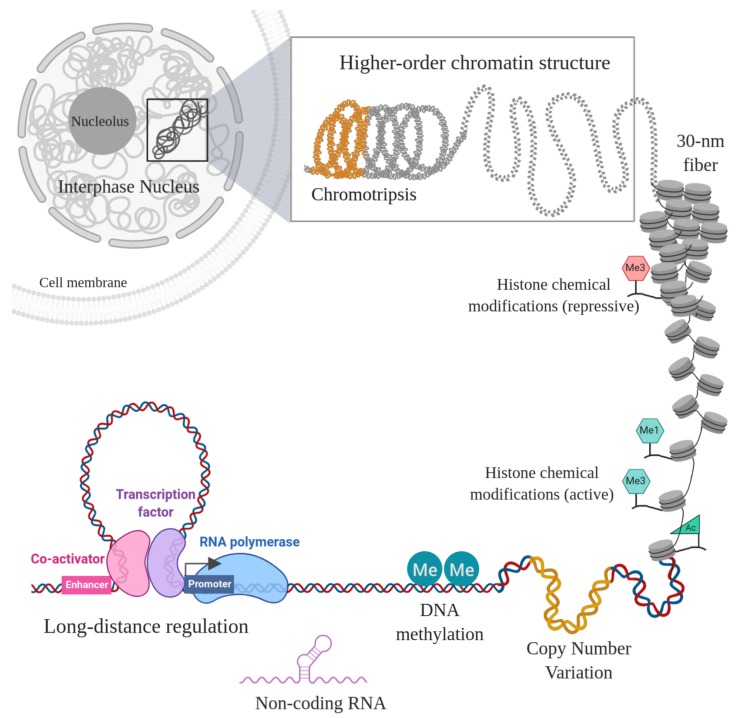
Gene expression regulatory mechanisms. Gene regulation in the eukaryotic cell nucleus involves different mechanisms that occur simultaneously at different spatial scales and biological contexts. These mechanisms include chromatin looping to allow contact between cis regulatory elements like enhancers and promoters, so recruitment of the transcriptional machinery can be facilitated; non coding RNA modulation of gene expression and silencing; methylation of DNA as an steric impediment to TFs binding, thus silencing transcription; and histone post transcriptional modifications that contribute to the electrostatic landscape of chromatin and encourage (e.g., H3K4me1 and H3K27ac that mark enhancer sequences or H3K4me3 that protects promoters from DNA methylation) or deter (e.g., H3K27me3 and H3K9me3 that mark heterochromatin) transcriptional processes. To date, the cooperativity and feedback between them remains to be fully characterized. (Image created with BioRender, https://biorender.com/).

**Figure 2 genes-10-00865-f002:**
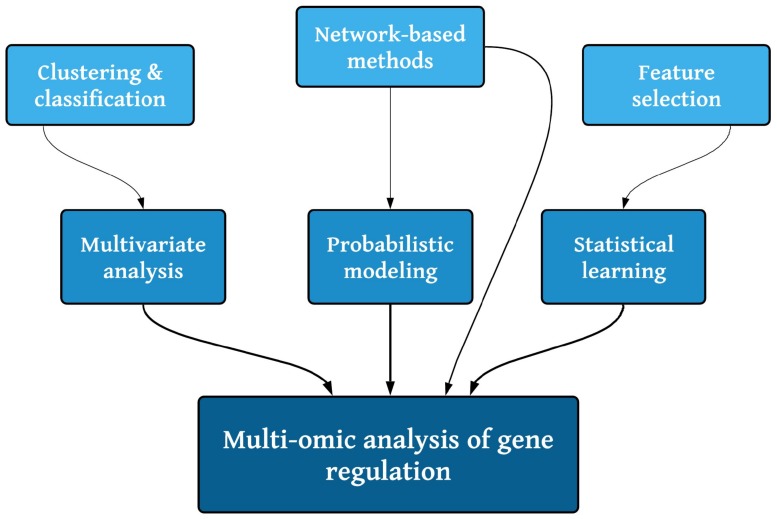
Several modeling approaches to integrate multiomic data in gene regulation.

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
