# Peer review of "The Many Faces of Gene Regulation in Cancer: A Computational Oncogenomics Outlook"

_genes, 2019, doi:10.3390/genes10110865_

Round 1

Reviewer 1 Report

The review is very comprehensive and a valuable contribution because it covers a large catalogue of techniques and methods used to study gene regulation on a genome-wide level. It also manages a good balance between details of selected methods and general broad view which makes the text easy to read.

The review could briefly mention some of the newer techniques that are becoming common practice for the study of gene regulation e.g., using the large catalogue of applications based on CRISPR to study the mechanisms of gene regulation.

Here are some comments and examples for typos and sentences that require some minor changes.

l82 sustenance

l102 methyl

l156 "characteristing is their length"
revise sentence

l168 "downregulation" of VIM, ZEB1, ...

p.10 l 327 introduce abbreviation DNMTs
p.11 l 328 introduce abbreviation MBD

p.11 l 340 "sex chromosomes"

l 344 For example: For the measurement of methylation levels the pipeline
outputs beta value statistics as a metric for each ... . M value statistics
explain, briefly why transformation is required.

l 480 "Similar network fusion" add citation [259]

l 489 box Omics integration stages (revise text in this section)
revise sentence "Major cancers from ovary, uterus, cervix ..."
revise sentence "Now that we know that some cases of uterine cancer ..."

revise sentence in new paragraph "To answer this sort of questions ..."

l529. revise sentence "The alpha parameter controls how similiar ..."

l538 should be more detailed. The problem for normalization of
data from different platform, metrics and measurement scales for
the application such as the presented linear models. In contrast to for example
methods such as Random Forests which are based on decision trees.

l544. replace "wins", e.g., when the largest sub-dataset dominates the combined data

l549 revise sentence: "Liu et al. noticed that ..."

l551 when generated from the combined dataset compared to the model generated for individual data types separately for simulated insilico data and cancer samples with paired gene expression and methylation data.

l554 revise sentence. "In conclusion, ..."
l555 revise sentence. "But stick out ..."

l578 "to found a set of genes" - to identify a set of genes
l579 "leading to find" e.g. that lead to the discovery of putative key regulatory elements in HNC.
Also revise sentence l579. "By means of these"

l602 expression

l609 hybrid

l612 revise sentence "Cancer is quite ..."

l613 cellular, 'tissue' and phenotypic 'level is'

Author Response

Authors’ response to the reviewers.

The authors are grateful to the three reviewers for their insightful and scholarly comments. Their suggestions and feedback have resulted in an important improvement on the readability and utility of our work. We have considered all their comments and concerns. Below we present a point-by-point response to all of them. To ease reading all of our comments and responses are in blue, bold print. The reviewer comments requiring a specific response appear in black, bold print whereas minor points such as typos appear in black print.

Reviewer 1

The review is very comprehensive and a valuable contribution because it covers a large catalogue of techniques and methods used to study gene regulation on a genome-wide level. It also manages a good balance between details of selected methods and general broad view which makes the text easy to read.

Response: The authors thank the reviewer for their positive comments on our manuscript.

The review could briefly mention some of the newer techniques that are becoming common practice for the study of gene regulation e.g., using the large catalogue of applications based on CRISPR to study the mechanisms of gene regulation.

Response:
A brief subsection about CRISPR-based approaches to study/modify gene regulation has been included in the revised manuscript (lines 482-519), as well as some key references (references 245-261).

Here are some comments and examples for typos and sentences that require some minor changes.

l82 sustenance

l102 methyl

l156 "characteristing is their length"

revise sentence

l168 "downregulation" of VIM, ZEB1, ...

p.10 l 327 introduce abbreviation DNMTs

p.11 l 328 introduce abbreviation MBD

p.11 l 340 "sex chromosomes"

l 344 For example: For the measurement of methylation levels the pipeline outputs beta value statistics as a metric for each ... . M value statistics explain, briefly why transformation is required.

Response: The use of beta value statistics is common since their interpretation is straightforward. However, since beta value distributions are often bimodal, it results convenient to transform to M values in order to carry out the differential methylation analysis using linear methods (e.g. limma) [see lines 345-350].

l 480 "Similar network fusion" add citation [259]

l 489 box Omics integration stages (revise text in this section)

revise sentence "Major cancers from ovary, uterus, cervix ..."

revise sentence "Now that we know that some cases of uterine cancer ..."

revise sentence in new paragraph "To answer this sort of questions ..."

l529. revise sentence "The alpha parameter controls how similar ..."

l538 should be more detailed. The problem for normalization of data from different platform, metrics and measurement scales for the application such as the presented linear models. In contrast to for example methods such as Random Forests which are based on decision trees.

Response: The manuscript has been revised to include a brief discussion in this regard [see discussion in lines 587-594 and 626-634].

l544. replace "wins", e.g., when the largest sub-dataset dominates the combined data

l549 revise sentence: "Liu et al. noticed that ..."

l551 when generated from the combined dataset compared to the model generated for individual data types separately for simulated in silico data and cancer samples with paired gene expression and methylation data.

l554 revise sentence. "In conclusion, ..."

l555 revise sentence. "But stick out ..."

l578 "to found a set of genes" - to identify a set of genes

l579 "leading to find" e.g. that lead to the discovery of putative key regulatory elements in HNC.

Also revise sentence l579. "By means of these"

l602 expression

l609 hybrid

l612 revise sentence "Cancer is quite ..."

l613 cellular, 'tissue' and phenotypic 'level is'

Response: The authors thank the reviewer for pointing out typos and grammar errors. All instances have been corrected.

Reviewer 2 Report

The authors provide a review (is it rather a perspective - as they state also in the text) about the wide area of "gene deregulation" in cancer. The review is written in clear language and easily understandable. They end with an account which computational models could/should be developed for making progress in the field of cancer gene deregulation.

Although the authors cover a wide range of concepts, their attempt to comprehesively cover (a) the literature of cancer-relevant genomic mechanisms, (b) space of experimental omics technolgies with relvance for cancer, (c) the space of possible solutions has to fail because the topic is too broad. This means that they can only be very superficial - still in the end ending up with 34 pages fraft mansucript. The "review" rather reads like an introduction of a Ph.D. thesis than a focused review. The aim of a review -to inform readers on a particular field of research and pointing to the most relevant primary articles- has not been met.

In particular, the final suggestions of computational modeling (Fig. 3) are very personal and should not be the conclusion of a review. It is debateable whether (or how) this approach is suited to integrate the complex data from diverse omics sources and help uncover known and new biological mechanisms.

Also, it would be helful if the authors highlight how expression-affecting cancer mechanisms manifest in all these sources of oncogenomic data and why new (their?) ways of analyzing data are doing better than traditional methods.

I cannot recommend the publication of this articla as a review in the present form. I recommend to  transform the manuscript into a pure review with better a deeper account on how cancerogenesis mechanisms manjifest itself in oncogenomic data and a clear link to the challenges in data analysis that this crerates.

Author Response

Authors’ response to the reviewers.

The authors are grateful to the three reviewers for their insightful and scholarly comments. Their suggestions and feedback have resulted in an important improvement on the readability and utility of our work. We have considered all their comments and concerns. Below we present a point-by-point response to all of them. To ease reading all of our comments and responses are in blue, bold print. The reviewer comments requiring a specific response appear in black, bold print whereas minor points such as typos appear in black print.

Reviewer 2

The authors provide a review (is it rather a perspective - as they state also in the text) about the wide area of "gene deregulation" in cancer. The review is written in clear language and easily understandable. They end with an account which computational models could/should be developed for making progress in the field of cancer gene deregulation.

Response: The authors would like to thank the reviewer for their positive comments about our manuscript.

Although the authors cover a wide range of concepts, their attempt to comprehensively cover (a) the literature of cancer-relevant genomic mechanisms, (b) space of experimental omics technologies with relevance for cancer, (c) the space of possible solutions has to fail because the topic is too broad. This means that they can only be very superficial - still in the end ending up with 34 pages draft manuscript. The "review" rather reads like an introduction of a Ph.D. thesis than a focused review. The aim of a review -to inform readers on a particular field of research and pointing to the most relevant primary articles- has not been met.

Response: The manuscript has been revised in order to better comply with the reviewer’s comments and suggestions. Particular effort has been made to highlight relevant issues, while at the same time being as self-contained as possible.

In particular, the final suggestions of computational modeling (Fig. 3) are very personal and should not be the conclusion of a review. It is debatable whether (or how) this approach is suited to integrate the complex data from diverse omics sources and help uncover known and new biological mechanisms.

Response: The revised version of the manuscript has been amended.  Personal and subjective statements related to our own research and views have been kept to a minimum.  The same is true about debatable or controversial matters and opinionated remarks.

Also, it would be helpful if the authors highlight how expression-affecting cancer mechanisms manifest in all these sources of oncogenomic data and why new (their?) ways of analyzing data are doing better than traditional methods.

Response: The manuscript has been revised to make evident some key aspects of cancer phenomenology that are associated with medium to large changes in gene expression. The need to use novel methods to analyze and integrate the data, along with more established approaches has also been explicitly stated in the revised manuscript [see in particular lines 49-53 and 713-717].

I cannot recommend the publication of this article as a review in the present form. I recommend to  transform the manuscript into a pure review with better a deeper account on how cancerogenesis mechanisms manifest itself in oncogenomic data and a clear link to the challenges in data analysis that this creates.

Response: Following the reviewer recommendations, the tone of the exposition has been redirected towards an informative and objective one. An effort has been made to present the different methodological approaches in a highly descriptive, neutral way so as to inform the readers along the lines of a comprehensive review rather than an opinionated piece.  The manuscript now includes a number of specific statements about how omic data derived from tumor samples is able to capture particular issues related to factors affecting gene regulation at a global transcriptomic level. We thank the reviewer for these suggestions and guidelines that helped us to better achieve our goal of communicating the important role of integration of multiple approaches to understand cancer-related gene deregulation.

Reviewer 3 Report

The review ”The many faces of gene regulation in cancer: A computational oncogenomics outlook” is a comprehensive and educational review paper. It reads well and is interesting to follow, although perhaps a little too long. Very nice with the combination of description of both work flow and data analysis. I think it would have been nice if the authors in the beginning clarified/defined what they in this review mean by “gene regulation” just to set the concrete context of the biological main topic.

Some sentences are very long and should be shortened or split in two, e.g. line 207-213; 305-311.

In section 2.1, “Established factors in gene regulation in cancer”, it would have been nice to have some more concrete cancer-relevant examples for each factor, e.g. for the first section on promoters and enhancers could have mentioned e.g. TERT promoter mutations leading to upregulation of telomerase expression observed in human cancer.

For the established vs. the emerging factors contributing to form the mechanistic basis for gene expression regulation (box on page 3) should not chromosome aberrations (and miRNAs) be listed as established factors (and not emerging as they have been studied and observed for quite some time)?

Challenges with validating results from modeling in independent cohorts due to overfitting (e.g. with LASSO) should be mentioned. It would be nice to have a paragraph on statistical power. Also, matrix factorization techniques applied to multi-omics data should be described (reviewed e.g. in https://www.sciencedirect.com/science/article/pii/S0168952518301240?via%3Dihub).

Line 70; “gene-coding” is a strange word, use another (the gene itself?). In line 71 it is a bit confusing how post-translational changes affect gene regulation (the topic of the review) – is it thought as in a feed-back loop with the protein acting on its “own” coding sequence or more in trans with post-translational modifications of TFs  that affect binding of target gene? Please clarify or remove.

Figure 1: this figure is a bit confusing as it looks likeDNA is “exported” out of the nucleus? Perhaps remove the lipid bilayer drawing?

As miRNAs may have both oncogenic and tumor-suppressive roles “oncogenic” in line 163 should be substituted with “oncogenic and tumor-suppressive”.

Minor comments – some spelling mistakes:

Line 52: usualy --> usually

Line 70: traslocations --> translocations

Line 98: trough --> through

Line 102 methy --> methyl

Line 156: lengt --> length

Line 180: the abbreviation for DNA methylatransferases should be “DNMTs” and not “MTFs” (see line 327)

Line 216: exon --> exome

Line 220: (germline) variation --> variants?

Line 607: “Panel B” --> “Panel A”?

Author Response

The authors are grateful to the three reviewers for their insightful and scholarly comments. Their suggestions and feedback have resulted in an important improvement on the readability and utility of our work. We have considered all their comments and concerns. Below we present a point-by-point response to all of them. To ease reading all of our comments and responses are in blue, bold print. The reviewer comments requiring a specific response appear in black, bold print whereas minor points such as typos appear in black print.

Reviewer 3

The review ”The many faces of gene regulation in cancer: A computational oncogenomics outlook” is a comprehensive and educational review paper. It reads well and is interesting to follow, although perhaps a little too long. Very nice with the combination of description of both work flow and data analysis. I think it would have been nice if the authors in the beginning clarified/defined what they in this review mean by “gene regulation” just to set the concrete context of the biological main topic.

Response: We would like to thank the reviewer for their comments and insightful remarks. To improve the clarity of presentation we have included a brief definition to set the scope of our study regarding what do we mean by “gene regulation” and “gene regulation programs” here by re-phrasing some paragraphs in section 2.

Some sentences are very long and should be shortened or split in two, e.g. line 207-213; 305-311.

Response: We have revised these instances and the whole manuscript, to avoid the use of long sentences.

In section 2.1, “Established factors in gene regulation in cancer”, it would have been nice to have some more concrete cancer-relevant examples for each factor, e.g. for the first section on promoters and enhancers could have mentioned e.g. TERT promoter mutations leading to upregulation of telomerase expression observed in human cancer.

Response: The revised version of the manuscript now includes a set of examples for many of the established mechanisms and processes. These examples are intended to point out to specific features affecting gene regulation (see, for instance lines 88-92).

For the established vs. the emerging factors contributing to form the mechanistic basis for gene expression regulation (box on page 3) should not chromosome aberrations (and miRNAs) be listed as established factors (and not emerging as they have been studied and observed for quite some time)?

Response: The manuscript has been revised accordingly with the suggestions of the reviewer. miRNAs and chromosome aberrations are now included in the established factor subsection.

Challenges with validating results from modeling in independent cohorts due to overfitting (e.g. with LASSO) should be mentioned. It would be nice to have a paragraph on statistical power. Also, matrix factorization techniques applied to multi-omics data should be described (reviewed e.g. in https://www.sciencedirect.com/science/article/pii/S0168952518301240?via%3Dihub).

Response: A brief paragraph discussing these matters has been included (see lines 587-594)

Line 70; “gene-coding” is a strange word, use another (the gene itself?). In line 71 it is a bit confusing how post-translational changes affect gene regulation (the topic of the review) – is it thought as in a feedback loop with the protein acting on its “own” coding sequence or more in trans with post-translational modifications of TFs  that affect binding of target gene? Please clarify or remove.

Response: We have removed the confusing statement

Figure 1: this figure is a bit confusing as it looks like DNA is “exported” out of the nucleus? Perhaps remove the lipid bilayer drawing?

Response: The figure has been redraw to clarify that there is a zooming process on display.

As miRNAs may have both oncogenic and tumor-suppressive roles “oncogenic” in line 163 should be substituted with “oncogenic and tumor-suppressive”.

Minor comments – some spelling mistakes:

Line 52: usualy --> usually

Line 70: traslocations --> translocations

Line 98: trough --> through

Line 102 methy --> methyl

Line 156: lengt --> length

Line 180: the abbreviation for DNA methylatransferases should be “DNMTs” and not “MTFs” (see line 327)

Line 216: exon --> exome

Line 220: (germline) variation --> variants?

Line 607: “Panel B” --> “Panel A”?

Response: The authors thank the reviewer for pointing out writing errors (typos and grammar mistakes). All instances have been corrected as suggested.

Round 2

Reviewer 2 Report

I still cannot recommend to publish. The scope of the ms did not change significantly

Author Response

Dear Reviewer 2, 

We do appreciate your view on our work, that although divergent from ours it help us to enrich the dialogue. We are the presenting you a respectful response as to why do we think that the broad scope of our review may be beneficial for certain readership in computational onco-genomics.

It is true that many review articles aim at being sharply focused so as to provide a close scholar view of a narrow, well-defined topic. Such a view is extremely useful, both as an introduction to the subject and as a report of the state of the art of a specific issue within the scientific sub-field in question.

It is also common (particularly in scientific disciplines such as physics, mathematics and computer science, but is an emerging trend in quantitative and computational biology) to write down longer review papers on much broader subjects. Such papers are by necessity (and by design!) not overly detailed and specific, but rather present a panoramic view supported with a vast, well-curated literature survey (for instance, our manuscript is admittedly long, however, around 14 and a half pages correspond to a list containing 308 detailed references). The purpose of these “Reports” is to introduce the reader to a broader subject, often an entire research subfield, touching upon the main features of interest, presenting the full set of matters from diverse standpoints and tracing a clear path to the literature for the readers interested in more precise and detailed accounts on the specifics. This is our goal with this manuscript. This goal borns out of necessity, given the wide, entangled nature of anomalous gene expression in cancer, its genome-wide experimental characterization and its computational analysis.

As is correctly noted by Reviewer 2, in our article we discuss the issue of ‘which computational models could/should be developed for making progress in the field of cancer gene deregulation’ . We aimed to do so at the light of the many disparate but converging biological phenomena involved, taking into account existing omic technologies to probe on the matters, as well as the intricacies, and limitations of the experimental approaches both individually and quite especially when combined to present a multi-omic description. We did so, to present a broad panorama of what are some of the challenges that the computational oncology analyst will face and, hopefully provide some clues.

Now, a question may arise as to how the broad scope of our review contributes or is relevant to the field of cancer genomics. Imagine for instance that a computational oncology research team is trying to analyze the epigenomics of prostate cancer on a large study group. Evidence suggests that both DNA methylation and small non-coding RNAs (esp. miRNAs) play important regulatory roles in prostate cancer so, a natural question would be ‘what is the concerted effect of both methylation and miRNA expression over anomalous expression on prostate tumors?’ To try to resolve this, it is likely that the team gathers RNASeq transcriptomic data from the tumors and controls, as well as methylation data (say from methylation arrays) and miRNA Seq data. All these types of data are pre-processed and analyzed with well established and robust computational algorithms and pipelines.

However, in order to build a model integrating all three effects, one has to consider the assumptions and limitations of each individual technique, the fact that their domains and dynamic ranges are different, but also the fact that different preprocessing assumptions may impact differently all of these types of data which in turn will have different effects depending the modeling approach for data integration chosen by such a research team. Admittedly it is quite a difficult task.

We believe that this task will become easier if the research team may refer to a document discussing (broadly but to a certain extent ‘superficially’) all these relevant issues and pointing them out to relevant sources in the literature for the parts they need to gain deeper understanding that will be different for every different team and project. This is the gap we are intending to fill with our broad review.

Academic editor comments:

Decision: Accept after minor revisions

AE 1. Notes for Authors: New techniques have been introduced. I suggest that more examples should be included for cancer omic studies. For example,  some other reviews (PMID: 24747696) and more examples which applied the technologies should be incorporated.   

Author's Reply: 

Dear Academic Editor

Our manuscript has been further revised to take into account the reviewers' and editor suggestions. The suggested review article (PMID: 24747696) and a couple of additional literature survey/perspective articles have been discussed and added to the reference list.

We have also re-revise the wording of some paragraphs to try to comply with some of Reviewer 2 initial suggestions in a better way that we did in the 1st revision. One thing however in which we respectfully differ from Reviewer 2 is in the actual scope of our review article. Our work has been always intended to be a 'broad review rather than a 'focused review'. We do not see this as some kind of defect or limitation of our manuscript but as a desired feature to to comply with our goal.

We have thus included the following response to the second round of comments from Reviewer 2:

Response: It is true that many review articles aim at being sharply focused so as to provide a close scholar view of a narrow, well-defined topic. Such a view is extremely useful, both as an introduction to the subject and as a report of the state of the art of a specific issue within the scientific sub-field in question. 

It is also common (particularly in scientific disciplines such as physics, mathematics and computer science, but is an emerging trend in quantitative and computational biology) to write down longer review papers on much broader subjects. Such papers are by necessity (and by design!) not overly detailed and specific, but rather present a panoramic view supported with a vast, well-curated literature survey (for instance, our manuscript is admittedly long, however, around 14 and a half  pages correspond to a list containing 308 detailed references). The purpose of these “Reports” is to introduce the reader to a broader subject, often an entire research subfield, touching upon the main features of interest, presenting the full set of matters from diverse standpoints and tracing a clear path to the literature for the readers interested in more precise and detailed accounts on the specifics. This is our goal with this manuscript. This goals borns out of necessity, given the wide, entangled nature of anomalous gene expression in cancer, its genome-wide experimental characterization and its computational analysis.

As is correctly noted by Reviewer 2, in our article we discuss the issue of ‘which computational models could/should be developed for making progress in the field of cancer gene deregulation’ . We aimed to do so at the light of the many disparate but converging biological phenomena involved, taking into account existing omic technologies to probe on the matters, as well as the intricacies, and limitations of the experimental approaches both individually and quite especially when combined to present a multi-omic description. We did so, to present a broad panorama of what are some of the challenges that the computational oncology analyst will face and, hopefully provide some clues.

Now, a question may arise as to how the broad scope of our review contributes is relevant to the field of cancer genomics. Imagine for instance that a computational oncology research team is trying to analyze the epigenomics of prostate cancer on a large study group. Evidence suggests that both DNA methylation and small non-coding RNAs (esp. miRNAs) play important regulatory roles in prostate cancer so, a natural question would be ‘what is the concerted effect of both methylation and miRNA expression over anomalous expression on prostate tumors?’ To try to resolve this, it is likely that the team gathers RNASeq transcriptomic data from the tumors and controls, as well as methylation data (say from methylation arrays) and miRNA Seq data. All these types of data are pre-processed and analyzed with well established and robust computational algorithms and pipelines.

However, in order to build a model integrating all three effects, one has to consider the assumptions and limitations of each individual technique, the fact that their domains and dynamic ranges are different, but also the fact that different preprocessing assumptions may impact differently all of these types of data which in turn will have different effects depending the modeling approach for data integration chosen by such a research team. Admittedly it is quite a difficult task. 

We believe that this task will become easier if the research team may refer to a document discussing (broadly but to a certain extent ‘superficially’) all these relevant issues and pointing them out to relevant sources in the literature for the parts they need to gain deeper understanding that will be different for every different team and project. This is the gap we are intending to fill with our broad review. 

Decision: Accept in current form

AE 2.

Decision: Accept after minor revisions

Notes for Authors Author's Reply: To ease reading, our response to Academic Editor will be presented for every item discussed (for this we have numbered the items)

1. I don’t really understand the organization of the paper, neither what is the rationale of including some very similar information inside the main text and other inside boxes.
i.e. concerning one company: Illumina. arrays are in the main text and sequencing in boxes
moreover, it is the only method for sequencing included

Response 1: The rationale to include some seemingly redundant information in the boxes was that these boxed sections are of a more procedural (methodological) nature and may be skipped on a first reading. However, following the advice of the Academic Editor (and to better comply with the journal’s style), we have eliminated the boxes and re-framed such content in the main text.

2. the paper contains a rather “artificial definition” of established and emerging gene regulation factors. i.e. why are chromosome aberrations established and chromosomal instability emerging?

Response 2: To avoid unnecessary and/or controversial comments and to improve readability, all regulation factors are discussed on a single section disregarding the “established vs emerging” definition that has been completely leaved out of this manuscript. 

3. I don’t understand much the logic behind the manuscript organization i.e.: what does the following sentence inside 3.1.2 RNA sequencing section mean?
" Because transcription must be tightly regulated to be specific, yet flexible, it is exquisitely controlled by a set of interconnected mechanisms comprising specific DNA sequences “cis elements” and specialized proteins “trans  elements” that operate in the nuclear epigenetic context to activate and repress genes in response to stimuli (i.e. signal transduction) or to manage developmental and differentiation processes"
The following paragraph inside the same section (RNA sequencing) do not much sense to me either: 
" Mutations that disrupt either cis-elements or the gene sequence of trans-elements involved in transcription can lead to dangerous deregulation of gene expression, which could eventually result in cancer…."

Response 3: The Editor is right; subsection 3.1.2 has been edited for clarity. The conflicting paragraphs have been edited out of this section.

4. In my opinion the paper is of potential interest but needs to be completely re-organized in a more comprehensive manner.

Response 4: Thank you for your comments and suggestions. We have re-organized sections, re-framed some paragraphs and get rid of text boxes by re-organizing the contents. Section re-organization was made in order to improve readability and comprehensiveness of the review. In particular, section 2 has been arranged into several subsections each one related to a single aspect of gene regulation. As stated previously, no distinction has been made as regards these factors. Section 3 has been re-framed into subsections related to different facets of the experimental techniques whereas such techniques as well as their experimental and analysis approaches are arranged in sub-subsections and annotated paragraphs. Section 4 has also been slightly re-organized to highlight how computational and statistical integration schemes arise.

AE 3.

Decision: Accept in current form

Notes for Authors. I believe that the paper is informative and acceptable, however, will greatly benefit of more figures of the type of Figure 1.  Response: Authors have desestimated the option to add additional figures than Figure 1-2 already on main text.